# Securing IoT Devices against Differential-Linear (DL) Attack Used on Serpent Algorithm

**Khumbelo Muthavhine \* and Mbuyu Sumbwanyambe \***

Department of Electrical and Mining Engineering, University of South Africa, Roodepoort, Johannesburg 1709, South Africa
\* Correspondence: kdmuthavhine@gmail.com (K.M.); sumbwm@unisa.ac.za (M.S.)

**Abstract:** Cryptographic algorithms installed on Internet of Things (IoT) devices suffer many attacks. Some of these attacks include the differential linear attack (DL). The DL attack depends on the computation of the probability of differential-linear characteristics, which yields a Differential-Linear Connectivity Table (*DLCT*). The *DLCT* is a probability table that provides an attacker many possibilities of guessing the cryptographic keys of any algorithm such as Serpent. In essence, the attacker firstly constructs a *DLCT* by using building blocks such as Substitution Boxes (S-Boxes) found in many algorithms' architectures. In depth, this study focuses on securing IoT devices against DL attacks used on Serpent algorithms by using three magic numbers mapped on a newly developed mathematical function called Blocker, which will be added on Serpent's infrastructure before being installed in IoT devices. The new S-Boxes with 32-bit output were generated to replace the original Serpent's S-Boxes with 4-bit output. The new S-Boxes were also inserted in Serpent's architecture. This novel approach of using magic numbers and the Blocker Function worked successfully in this study. The results demonstrated an algorithm for which its S-Box is composed of a 4-bit-output that is more vulnerable to being attacked than an algorithm in which its S-Box comprises 32-bit outputs. The novel approach of using a Blocker, developed by three magic numbers and 32-bits output S-Boxes, successfully blocked the construction of *DLCT* and DL attacks. This approach managed to secure the Serpent algorithm installed on IoT devices against DL attacks.

**Keywords:** security of IoT devices; serpent; differential-linear attack (DL attack); differential-linear connectivity table (*DLCT*); magic numbers

## 1. Introduction

The IoT has seamlessly woven itself into people's lives based on the fact that everyone is finding the technology useful in terms of the support that accompanies it. This support is not limited to making the lives of people easier [1–3]. For example, the connection from intelligent thermostats, home hubs, remote door locks, and numerous app-controlled devices has made the lives of the people much more interesting and and of high quality, both for manufacturing and daily use. It is becoming more and more satisfying in people's lives in various ways [4].

IoT supports users in working smarter, living more innovatively, and achieving total control over users' lives [3–5]. In addition to users' smart home devices, IoT is an indispensable technology in trade and industry in providing companies a real-time glimpse into the internal operations of the company's practices [5]. IoT provides insights into everything from machine production to supply chain and logistics operations (this is from the warehouse level to the customer's door) [3,4]. IoT enables businesses to automate methods and save capital on employment. Moreover, IoT lessens waste and enhances service delivery by rendering production and delivery of products less expensive and rendering transparency into client transactions [5]. IoT empowers organizations to reduce expenses, increase security, and enhance quality from end to end, which tranlates to a

win-win situation for both clients and suppliers [4,5]. Even though IoT is beneficial for the community and manufacturers, there are difficulties related to the deployment of the IoT, such as privacy and security of private data.

The privacy difficulties postured by the IoT are related to those postured by existing digital technologies that obtain and transmit data, especially radio-frequency identification and cloud computing. IoT devices are everywhere, and users have little experience on the know-how involved in managing data [4].

When executing an action that depends on digital technologies, including the IoT, users should consider the potential budgetary and social importance of possible digital protection occurrences concerning availability, integrity, or data confidentiality in data operation [3]. These values can weaken resources (for instance, through the interruption of transactions), threaten reputation (for example, through the disclosure of private data or website damage), or modify the business environment (for instance, through deprivation of innovation) [3–5].

Privacy encompasses the practices on which personal data can be obtained and is, consequently, one of the most prominent challenges. The processes of tracing, verification, validation of devices, all activities performed, and collecting private data from different forms can foster an environment for effortless hacking with respect to information [5].

Security is one of the common difficulties that the IoT has to address [3]. Affordability and low expense broadband connection and Wi-Fi abilities in numerous devices are required for secure localization in common areas, and transmitting them unprotected would yield them to cyber-attacks [3,5]. IoT enables consistent data sharing between similar gadgets and distinguishes three principal components guaranteeing security—authentication, access control, and confidentiality of IoT [4]. A robust cryptographic algorithm is needed to secure data collected, used, stored, and transmitted using IoT devices [3]. IoT devices depend on cryptographic algorithms to store and transmit confidential information [6–9]. While the improvements of security on IoT devices are increasingly developing using vigorous cryptographic algorithms, more attackers develop various methods of attacking the notably strong algorithms [9].

The Serpent algorithm is one of the most popular algorithms and is usually installed on IoT devices. However there are concerns regarding the robustness of the Serpent algorithm in terms of security. One of the main concerns is the vulnerability of Serpent algorithms towards *DLCT*. *DLCT* was used to attack Serpent algorithms so as to discover secret encryption keys [6,7]. Once an invader cracks and discovers the Serpent's key, all data encrypted with Serpent in IoT devices can be easily obtained by the attackers. This attack can result in the exploitation of entire IoT devices and their users. Essentially, it is easy for an attacker to attack a cryptographic algorithm such as Serpent because it has a low number of output bits (4-bits) found on the S-Boxes [8]. The Differential-Linear attack can harm the entire security system of IoT devices if it is not appropriately protected against. A few studies have been conducted to secure the Serpent algorithm against DL attacks. This study focuses on strengthening Serpent from DL attacks by constructing a new additional function called Blocker. This is performed by using three magic numbers and developing the new 32-bit output S-Boxes so that *DLCT* becomes cumbersome when building *DLCT* using the S-Boxes of the Serpent algorithm. It has been analyzed that DL attacks start with *DLCT*. Therefore, by blocking the construction of *DLCT*, it is believed that a DL attack will be impossible.

The *DLCT* can be prevented by using a blocker with three magic numbers. The first magic number is $Q = 4,302,746,963$, the second is $P = 4,559,351,687$, and the third is $M = 4,294,967,296$ mapped on Blocker, which will be inserted on Serpent's architecture. New 32-bit output S-Boxes were generated to replace the original 4-bits output S-Boxes. In this study, newly generated 32-bits output, S-Boxes, and Blocker Function successfully managed to secure the Serpent algorithm by blocking the construction of a probability table called *DLCT* used during the process of DL attack.

Additionally, a newly generated function called Blocker is introduced in this study. A Blocker Function uses a 32-bit output value from S-Box as *state*32*hold* and delivers a new value *statehold* value as an output. A Blocker Function also transforms a *P* value, *M* value, and *Q* value into unfactorizable polynomials. Random numbers and XOR operators are utilized for complexity and confusion in order to prevent reverse engineering for attackers. The XOR operator and *rand*() change the values of the variables inside a Blocker Function. The random numbers and XOR operators also provide a problematic input range when invaders reverse back a Blocker Function to calculate the exact information utilized. The value of *M*, *P*, and *Q* are also continually maintained as unfactorizable polynomial variables, which are non-linear and hard to reverse, in order to construct *DLCT* using any machine or computer. The random numbers and XOR operators include inventing hidden, unseen, and unchangeable variables for intruders. A Blocker Function produces a unique 32-bit S-Box suitable for the new Magic Serpent Algorithm. A Blocker Function distracts the attacker since it has many mathematical random numbers and XOR operators. Additionally, most mathematical XOR operators and random numbers are irreversible. For more mathematical characteristics of a Blocker Function and C++ explanations, refer to Section 4 and Figure 1.

```cpp
uint8_t Blocker(uint64_t state32hold, uint8_t statehold)
  //The variable state32hold is an
 //output of a newly 32-output-bit generated S-Box,
  //At this stage, state32hold is an input of Blocker.
  // statehold is an output of Blocker.
{   uint64_t Q=0, P=0, TempState =0, M=4294967296;
    uint64_t iSecret;
    state32hold = state32hold * (bool)(state32hold/ M)
    + M * (bool)(M / state32hold);
    if (state32hold > M)
    {   Q = 4559351687;
        P= 4302746963 % Q;
//M-value, P-value and Q value are unfacorizable polynomials.
        iSecret = rand() % (P^Q);
//Intializing a random number to be used in Blocker.
    }
    else
    {//Unfactorizable polynomial numbers are
     //non-linear and cumbersome to construct LAT tables using PC.
        iSecret = rand()% (Q^M);
        M = state32hold;
        Q = state32hold <<2;
        P = state32hold <<4;
        Q = M ^ P;
        P = M ^ Q;
        M = Q ^ P;
//Changing M, P and Q to be random of a range iSecret
        M = rand()% iSecret;
        P = rand()% iSecret;
        Q = rand()% iSecret;
//Note that iSecret is also random from iSecret = rand()% (Q^M);
//At this stage M, P and Q are unknown, invisible and irreversible to
//an intruder, since they are random numbers.
    }while (Q % M)    //Second modular operator (%) to make Q-value
        {    //unknown invisible and irreversible to an intruder.
        TempState = (~state32hold) & Q;
        state32hold = _abs64(state32hold^ Q);
        statehold= ((state32hold)/P)^0273;
        iSecret = rand()% (P^M);
        Q = TempState << 1;
        }
//Changing M, P and Q to be random of a range statehold
        Q = rand()% statehold;
        M = rand()% statehold;
        P = rand()% statehold;
        return statehold;
//At this stage M, P and Q are unknown, invisible and irreversible
//to an intruder, since they random.
}
```

**Figure 1.** New generated function called Blocker.

*1.1. Serpent Algorithm*

The Serpent is the cryptographic algorithm, a block cipher that encrypts and decrypts a data block of 128-bits using different sizes of the keys, such as 128, 192, and 256-bits [10]. The Serpent has three main building blocks. The building blocks are the mathematical functions used in the construction of an algorithm. These three main blocks are described as follows:

1.  Initial Permutation denoted by $IP$. The function of $IP$ is to rearrange an original order of the plaintext before the encryption process using Equation (1) where $Original Plaintext$ is the input of $IP$. A symbol of $"*"$ is a multiplication operator. $OutputIP$ is an output of $IP$, and $mod(127)$ is a mathematical modulus of 127. Refer to Equation (1).

$$OutputIP = (Original Plaintext * 32)mod(127) \tag{1}$$

2.  Serpent has a 32-round function composed of subkeys (key mixing), eight S-Boxes, and a linear transformation. The 32-round function is mathematically explained by a mathematical expression provided in Figure 2.

The linear transformation $L$ is the application of a series of rotations ($<<<$), shifts ($<<$) and XORs ($\oplus$) between the words of the current internal state. Let $W_0, W_1, W_2$ and $W_3$ be the 4 words

The following operations are done sequentially:

$W_0 <<< 13$. $W_2 = W_2 <<< 3$.

$W_1 = W_1 \oplus W_0 \oplus W_2$.

$W_3 = W_3 \oplus W_2 \oplus (W_0 << 3)$.

$W_1 = W_1 <<< 1$. $W_3 = W_3 <<< 7$.

$W_0 = W_0 \oplus W_1 \oplus W_3$.

$W_2 = W_2 \oplus W_3 \oplus (W_1 << 7)$.

$W_0 = W_0 <<< 5$. Finally, we have

$W_2 = W_2 <<< 22$.

**Figure 2.** 32-round function of Serpent.

3.  Serpent has final permutation $IP^{-1}$ function, which is an inverse of initial permutation $IP$.

Serpent uses eight $4 \times 4$ S-Boxes during the encryption process. These S-Boxes, together with their inverses, are defined in Tables 1–8. For instance, if the input of $S_0(X)$ is $0 = X$, then the output is 3, and $S_0(0) = 3$. If the input of $S_1(X)$ is $1 = X$, then the output is 12 and $S_1(1) = 12$. If the input of $S_7(X)$ is $2 = X$, then the output is 15, $S_7(1) = 15$, and so on. The same applies to the inverse cases. Refer to Figure 2. Serpent requires 33 of 128-bits, and subkeys are generated from an original key given by the user before encryption starts. The user can provide an original key size of 128, 192, or 256-bits long. In this study, the original 128-bit key is used to demonstrate how other 33-bit subkeys are generated using the mathematical expression given in Figure 3.

**Table 1.** First S-Box of Serpent defined as $S_0(X)$.

| $X$ | 0 | 1 | 2 | 3 | 4 | 5 | 6 | 7 | 8 | 9 | A | B | C | D | E | F |
|---|---|---|---|---|---|---|---|---|---|---|---|---|---|---|---|---|
| $S_0(X)$ | 3 | 8 | F | 1 | A | 6 | 5 | B | E | D | 4 | 2 | 7 | 0 | 9 | C |
| $InvS_0(X)$ | D | 3 | B | 0 | A | 6 | 5 | C | 1 | 4 | 4 | 7 | F | 9 | 8 | 2 |

**Table 2.** Second S-Box of Serpent defined as $S_1(X)$.

| $X$ | 0 | 1 | 2 | 3 | 4 | 5 | 6 | 7 | 8 | 9 | A | B | C | D | E | F |
|---|---|---|---|---|---|---|---|---|---|---|---|---|---|---|---|---|
| $S_1(X)$ | F | C | 2 | 7 | 9 | 0 | 5 | A | 1 | B | E | 8 | 6 | D | 3 | 4 |
| $InvS_1(X)$ | 5 | 8 | 2 | E | F | 6 | C | 3 | B | 4 | 7 | 9 | 1 | D | A | 0 |

**Table 3.** Third S-Box of Serpent defined as $S_2(X)$.

| $X$ | 0 | 1 | 2 | 3 | 4 | 5 | 6 | 7 | 8 | 9 | A | B | C | D | E | F |
|---|---|---|---|---|---|---|---|---|---|---|---|---|---|---|---|---|
| $S_2(X)$ | 8 | 6 | 7 | 9 | 3 | C | A | E | C | 1 | E | 4 | 0 | B | 5 | 2 |
| $InvS_2(X)$ | C | 9 | F | 4 | B | C | 1 | 2 | 0 | 3 | 6 | D | 5 | 8 | A | 7 |

**Table 4.** Fourth S-Box of Serpent defined as $S_3(X)$.

| $X$ | 0 | 1 | 2 | 3 | 4 | 5 | 6 | 7 | 8 | 9 | A | B | C | D | E | F |
|---|---|---|---|---|---|---|---|---|---|---|---|---|---|---|---|---|
| $S_3(X)$ | 0 | F | B | 8 | C | 9 | 6 | 3 | D | 1 | 2 | 4 | A | 7 | 5 | E |
| $InvS_3(X)$ | 0 | 9 | A | 7 | B | E | 6 | D | 3 | 5 | B | 2 | 4 | 8 | F | 1 |

**Table 5.** Firth S-Box of Serpent defined as $S_4(X)$.

| $X$ | 0 | 1 | 2 | 3 | 4 | 5 | 6 | 7 | 8 | 9 | A | B | C | D | E | F |
|---|---|---|---|---|---|---|---|---|---|---|---|---|---|---|---|---|
| $S_4(X)$ | 1 | F | 8 | 3 | C | 0 | B | 6 | 2 | 5 | 4 | A | 9 | E | 7 | D |
| $InvS_4(X)$ | 5 | 0 | 8 | 3 | A | 9 | 7 | E | 2 | C | B | 6 | 4 | F | D | 1 |

**Table 6.** Sixth S-Box of Serpent defined as $S_5(X)$.

| $X$ | 0 | 1 | 2 | 3 | 4 | 5 | 6 | 7 | 8 | 9 | A | B | C | D | E | F |
|---|---|---|---|---|---|---|---|---|---|---|---|---|---|---|---|---|
| $S_5(X)$ | F | 5 | 2 | B | 4 | A | 9 | C | 0 | 3 | E | 8 | D | 6 | 7 | 1 |
| $InvS_5(X)$ | 8 | F | 2 | 9 | 4 | 1 | D | E | B | 6 | 5 | 3 | 7 | C | B | 0 |

**Table 7.** Seventh S-Box of Serpent defined as $S_6(X)$.

| $X$ | 0 | 1 | 2 | 3 | 4 | 5 | 6 | 7 | 8 | 9 | A | B | C | D | E | F |
|---|---|---|---|---|---|---|---|---|---|---|---|---|---|---|---|---|
| $S_6(X)$ | 7 | 2 | C | 5 | 8 | 4 | 6 | B | E | 9 | 1 | F | D | 3 | A | 0 |
| $InvS_6(X)$ | F | A | 1 | D | 5 | 3 | 6 | 0 | 4 | 9 | E | 7 | 2 | C | 8 | B |

**Table 8.** Eighth S-Box of Serpent defined as $S_7(X)$.

| $X$ | 0 | 1 | 2 | 3 | 4 | 5 | 6 | 7 | 8 | 9 | A | B | C | D | E | F |
|---|---|---|---|---|---|---|---|---|---|---|---|---|---|---|---|---|
| $S_7(X)$ | 1 | C | F | 0 | E | 8 | 2 | B | 7 | 4 | C | A | 9 | 3 | 5 | 6 |
| $InvS_7(X)$ | 3 | 0 | 6 | D | 9 | E | F | 8 | 5 | C | B | 7 | A | 1 | 4 | 2 |

It receives the 128, 192 or 256-bit secret key as input and generates 33 subkeys with 128 bits each. The input has 8 words indexed from $w_{-8}$ to $w_{-1}$.

Then, the *pre-key* is calculated, which are 132 words indexed from $w_0$ to $w_{131}$ in the following manner:

$$w_i = (w_{i-8} \oplus w_{i-5} \oplus w_{i-3} \oplus w_{i-1} \oplus i \oplus \phi) << 11$$

where $\phi$ is the golden ratio

$$(\sqrt{5}+1)/2 \text{ or } 0x9e3779b9$$

From the pre-key we generate the 133 words of the subkeys.

Each word can be written as

$$k_i = SBox_{(3-(i \bmod 33)) \bmod 32}(w_i).$$
$$0 \leq i \leq 32.$$

**Figure 3.** Key generation of Serpent.

*1.2. Differential-Linear Attack*

The differential-linear attack is the mathematical procedure that is used in attacking algorithms by constructing a probability table called *DLCT* using S-Boxes in order to guess the keys [6,7]. An attacker chooses input pairs ($P_1$ and $P_2$) of an S-Box and analyzes the output pairs ($C_1$ and $C_2$) to construct *DLCT* using Equation (2). From Equation (2), there is $\Delta$, which is calculated as $P_1 \oplus P_2$ and $\lambda$, which is calculated as $C_1 \oplus C_2$. Multiplication is calculated using the dot multiplication operator to indicate that bits are multiplied instead of entire bytes.

$$DLCT(\Delta, \lambda) = \sum_{S_i(x)\epsilon[1,0]} (-1)^{\lambda \cdot (S_i(x) \oplus S_i(x \oplus \Delta))} \tag{2}$$

It is already stated that Equation (2) is used to construct *DLCT* using S-Box: For instance, if the first S-Box of Serpent is defined by Table 1, which has 4-bits input and 4-bits output chosen, then the *DLCT* will be a $2^4 \times 2^4$ matrix. Generally, if an S-Box has N-bits of input and M-bits of output, then its *DLCT*, when constructed, will be a $2^N \times 2^M$ matrix. Hence, the *DLCT* of the first S-Box of Serpent defined in Table 1 is said to be $2^4 \times 2^4$. With the aid of Equation (2), the *DLCT* of the first S-Box of Serpent is constructed and given in Table 9. With the aid of Table 9, an attacker can guess the key statistically by using probability theory. The highest number is eight in Table 9. The probability of guessing a key is 8/16, which is approximately the probability of guessing the head side when a coin is tossed. In simple terms, it is easy for an attacker to attack an algorithm with the aid of *DLCT*. The attacker checks the correlation between $C_1 \cdot \lambda$ and $C_2 \cdot \lambda$, if the correlation is high, then the key is discovered by using *DLCT*. While the elemental application of *DLCT* is for discovering a more accurate key investigation of the DL attack, it can be applied to improve DL attacks to the next advanced level. This study proved that attackers can apply the *DLCT* to choose the differential for $C_1$ and the linear approximation for $C_2$ in a manner that exposes the correlation between $C_1$ and $C_2$ to the attackers' advantage [6].

**Table 9.** The *DLCT* of the Serpent's first S-Box $S_0(X)$.

| Δ\λ | 0 | 1 | 2 | 3 | 4 | 5 | 6 | 7 | 8 | 9 | A | B | C | D | E | F |
|---|---|---|---|---|---|---|---|---|---|---|---|---|---|---|---|---|
| 0 | 8 | 8 | 8 | 8 | 8 | 8 | 8 | 8 | 8 | 8 | 8 | 8 | 8 | 8 | 8 | 8 |
| 1 | 8 | 0 | −4 | 0 | −4 | −4 | 0 | 4 | 0 | −4 | 0 | 0 | 0 | 4 | 0 | 0 |
| 2 | 8 | 0 | 0 | 0 | −4 | 0 | 0 | −4 | −8 | 0 | 0 | 0 | 4 | 0 | 0 | 4 |
| 3 | 8 | −4 | 0 | 0 | 4 | −4 | 0 | −4 | 0 | 0 | −4 | 0 | 0 | 4 | 0 | 0 |
| 4 | 8 | 0 | 0 | −8 | 0 | 0 | 0 | 0 | −8 | 0 | 0 | 8 | 0 | 0 | 0 | 0 |
| 5 | 8 | 4 | 0 | 0 | 0 | 0 | −4 | 0 | 0 | 0 | 4 | 0 | −4 | 0 | −4 | −4 |
| 6 | 8 | −4 | −4 | 0 | 0 | 0 | 0 | 0 | 8 | −4 | −4 | 0 | 0 | 0 | 0 | 0 |
| 7 | 8 | 0 | 4 | 0 | 0 | 0 | −4 | 0 | 0 | 4 | 0 | 0 | −4 | 0 | −4 | −4 |
| 8 | 8 | −4 | 0 | 0 | −4 | 0 | −4 | 4 | 0 | 0 | −4 | 0 | 0 | 0 | 4 | 0 |
| 9 | 8 | 0 | 0 | −8 | 0 | 0 | 0 | 0 | 0 | 0 | 0 | 0 | 0 | 0 | 0 | 0 |
| A | 8 | 0 | −4 | 0 | 4 | 0 | −4 | −4 | 0 | −4 | 0 | 0 | 0 | 0 | 4 | 0 |
| B | 8 | 0 | 0 | 0 | −4 | 0 | 0 | −4 | 0 | 0 | 0 | −8 | 4 | 0 | 0 | 4 |
| C | 8 | 0 | 4 | 0 | 0 | −4 | 0 | 0 | 0 | 4 | 0 | 0 | −4 | −4 | 0 | −4 |
| D | 8 | −4 | −4 | 8 | 0 | 4 | 4 | 0 | 0 | −4 | −4 | 0 | 0 | −4 | −4 | 0 |
| E | 8 | 4 | 0 | 0 | 0 | −4 | 0 | 0 | 0 | 0 | 4 | 0 | −4 | −4 | 0 | −4 |
| F | 8 | 0 | 0 | 0 | 0 | 4 | 4 | 0 | 0 | 0 | 0 | −8 | 0 | −4 | −4 | 0 |

*1.3. The Magic Number*

The magic number refers to the anti-design of using a constant integer directly to a source code of an algorithm. The magic number is applied to break one of the oldest functionality of coding [2]. The magic number renders the source code more cumbersome with respect to being modified and analyzed by an attacker [11]. Magic numbers are more confusing to an attacker when the same constant is applied to one section of an algorithm's source code without the derivative [2,11].

*1.4. Objective of the Study*

The Serpent algorithm is one of the typical traditional algorithms installed on IoT devices (example smart cards, sensors, remote controls, and intelligent cameras). The main problem is the *DLCT*, which is used in Serpent to reveal hidden encryption keys by intruders utilized to secure data stored in the IoT devices. In this study, a newly generated 32-bit output S-Boxes and a Blocker Function is proposed to ensure that the Serpent algorithm is protected from DL attacks, and the development of a feasibility table called *DLCT* employed during the process of a DL attack has to be blocked. A proposed Blocker Function uses a 32-bit output value from S-Box as *state*32*hold* and provides a new *statehold* value as an output. A Blocker Function also offers *P* value, *M* value, and *Q* value as unfactorizable polynomials. Random numbers and XOR operators are used for complexity and confusion in order to stop reverse engineering used by intruders. The random numbers and XOR operators are also proposed in order to provide a problematic input range when invaders reverse a Blocker Function to calculate the exact information utilized in that situation. For more mathematical characteristics of a Blocker Function and C++ explanations, refer to Figure 1. For more detail of a Blocker Function and flowchart, refer to Appendix A Figure A1.

*1.5. The Numerous DL Attacks on Serpent Algorithm*

Anderson et al. [12] attacked the Serpent algorithm using a DL attack and *DLCT* table. Eighty-six percent of key bits were discovered. Compton et al. [13] developed a Simple Power Analysis attack (SPA) to attack an 8-bit smart card encrypted by Serpent. The results showed

that Serpent key generation was convincing to a side-channel attack because of a linear feedback shift register (LFSR). LFSRs were very common in most cryptographic algorithms; suggestions were given that Serpent's LFSRs should be carefully modified and guesstimated in order to reduce attacks. Bar-On et al. [6] developed a new tool called *DLCT* used to attack Serpent's secret keys. Canteaut et al. [7] analyzed the observation of *DLCT* to obtain absolute indicators of Serpent weaknesses. Canteaut et al. [7] expanded the analytic results found on *DLCT* and DL attacks. Canteaut et al. [7] improved the observations about the notion of *DLCT* and DL attacks. According to the results found by Canteaut et al. [7], the *DLCT* approach method was found to be similar to the auto-correlation spectrum entities, and a conclusion was drawn that *DLCT* was nothing else but an Auto Correlation Table (ACT). Furthermore, Canteaut et al. [7] indicated that the ACT spectrum was invariable under any equivalence similarities and was not invariant under changes. Biham et al. [14] attacked the Serpent algorithm using the DL attack with the aid of the *DLCT* tool. Therefore, there is no denial that the Serpent algorithm is attackable by using DL attacks and the *DLCT* table. For more information on Serpent attacks, refer to the literature review of this study.

## 2. Problem Statement

The Serpent algorithm is one of the most common algorithms required to be installed on IoT devices. The main concern is a new tool called *DLCT* used to attack Serpent by intruders in order to discover secret encryption keys used to secure data stored in IoT devices. The process of using *DLCT* to find the key is called a DL attack [6,7]. Once an attacker cracks and discovers Serpent's key, all data encrypted with Serpent in IoT devices can be easily accessible to attackers. This attack can result in the exploitation of entire IoT devices and their users. Essentially, it is simple for an attacker to attack a cryptographic algorithm such as Serpent, since it has a low number of output bits (4-bit) found on the S-Boxes [8]. A differential-linear attack can harm the entire security of IoT devices if it is not appropriately considered. Little has been conducted to secure the Serpent algorithm against DL attacks. This study focuses on securing Serpent from DL attacks by constructing a new additional function called Blocker, using three magic numbers, and developing new 32-bit output S-Boxes so that *DLCT* will be cumbersome for building *DLCT* using the S-Boxes of the Serpent algorithm. It has been analyzed that DL attack starts with *DLCT*. Therefore, by blocking the construction of *DLCT*, it is believed that a DL attack will be impossible.

Additionally, a newly generated function called Blocker is introduced in this study. A Blocker Function uses a 32-bit output value from S-Box as *state32hold* and delivers a new value, *statehold* value, as an output. A Blocker Function also turns *P* value, *M* value, and *Q* value into unfactorizable polynomials. Random numbers and XOR operators are utilized for complexity and confusion in order to prevent reverse engineering for attackers. The XOR operator and *rand*() change the values of the variables inside a Blocker Function. The random numbers and XOR operators also provide a problematic input range when invaders reverse a Blocker Function to calculate the exact information utilized in that situation. The values of *M*, *P*, and *Q* are also continually maintained as unfactorizable polynomial variables, which are non-linear and hard to reverse, in order to construct *DLCT* using any machine or computer. The random numbers and XOR operators include inventing hidden, unseen, and unchangeable variables for intruders. A Blocker Function produces a unique 32-bit S-Box suitable for the new Magic Serpent Algorithm. A Blocker Function distracts the attacker since it comprises many mathematical random numbers and XOR operators. Additionally, most mathematical XOR operators and random numbers are irreversible. For more mathematical characteristics of a Blocker Function and C++ explanations, refer to Section 4 and Figure 1.

## 3. Literature Review

Biham et al. [14] developed a Serpent as an algorithm to replace the Data Encryption Standard (DES) algorithm. The main purpose of developing Serpent was to increase the Avalanche Effect (AE) in order to confuse and frustrate attackers [15].

Muthavhine and Sumbwanyambe [16] indicated that Serpent had been one of the most cryptographic algorithms used on IoT devices. In addition, Anderson et al. [12] demonstrated that Serpent had been used on IoT devices such as smart cards, Intel Pentium, and other 8-bit processors.

Sehrawat and Gill [17] indicated that nowadays, IoT devices such as smart cards play a critical function in everyone's life by delivering excellent services to the facet of the cyber world. IoT devices could also provide services such as intelligent management and monitoring. Additionally, Sehrawat and Gill [17] indicated that there has been an increase in 5G network dependence for seamless services; IoT devices had been attracting much attention to researchers. However, IoT devices in this miscellaneous 5G network are vulnerable to many attacks. It was already stated that most IoT devices encrypt data using Serpent.

Tezcan and Ozbudak [18] tried to reduce attacks found on Serpent by using differential factors. The differential factor attacked the key size using differential cryptanalysis and time complexity. Dunkelman et al. [19] developed more accurate results of the DL attack on round number 11 of Serpent found on IoT devices. The results were found using statistical analysis, mathematical theory, and experimental criticisms, which showed and declared that early attacks had exaggerated effects. Compton et al. [13] developed a Simple Power Analysis attack (SPA) to attack an 8-bit smart card encrypted by Serpent. The results showed that Serpent key generation was convincing to a side-channel attack because of a linear feedback shift register (LFSR). LFSRs are very common in most cryptographic algorithms; suggestions were given that Serpent's LFSRs should be carefully modified and guesstimated in order to reduce attacks.

Biham et al. [20] presented a linear approximation on round number nine of Serpent using a statistical theory of $1/2 + 2^{-52}$ probability. Furthermore, Biham et al. [20] continued using the theory to attack round number 10 of Serpent using all key sizes: data complexity of $2^{118}$ and time taken of $2^{89}$ seconds. A random variable of the probability was also applied on the first attack against round number 11 of Serpent using 192-bit and 256-bit key lengths, which needed an equal quantity of data and $2^{187}$ seconds taken [20].

Bar-On et al. [6] developed a *DLCT* used to attack Serpent's secret keys. Canteaut et al. [7] analyzed and observed a *DLCT* to obtain absolute indicators of Serpent weaknesses. Canteaut et al. [7] expanded the analytic results found on the *DLCT* and DL attacks. Canteaut et al. [7] improved the observations about the notion of *DLCT* and DL attack. According to the results found by Canteaut et al. [7], the *DLCT* approach method was found to be similar with respect to auto-correlation spectrum entities, and a conclusion was drawn that *DLCT* was nothing else but an Auto Correlation Table (ACT). Furthermore, Canteaut et al. [7] indicated that the ACT spectrum was invariable under any equivalence similarities and was not invariant under changes.

Bar-On et al. [6] indicated that the *DLCT* was formulated expeditiously using the fast Fourier transform. Additionally, Bar-On et al. [6] applied the *DLCT*'s strength to enhance DL attacks on ICEPOLE and DES in order to justify published experimental findings on CAESAR and Serpent. The results showed that *DLCT* was not abided by the DL attack model. Little has been conducted with respect to securing Serpent against DL attacks. This study focuses on securing Serpent, found on IoT devices, against DL attacks by using new 32-bit S-Boxes and a new function developed from three magic numbers in such a way that *DLCT* will be cumbersome to construct when using new 32-bit S-Boxes. During the study, the analysis of the results showed that DL started with *DLCT*. Therefore, by blocking the construction of *DLCT*, it was believed that the DL attacks would be impossible.

Additionally, a newly generated function called Blocker is introduced in this study. A Blocker Function uses a 32-bit output values from S-Box as *state*32*hold* and delivers a new value, *statehold*, as an output. A Blocker Function also turns *P* value, *M* value, and *Q* value into unfactorizable polynomials. Random numbers and XOR operators are utilized for complexity and confusion in order to prevent reverse engineering for attackers. The XOR operator and *rand*() change the values of the variables inside a Blocker Function.

The random numbers and XOR operators also provide a problematic input range when invaders reverse a Blocker Function to calculate the exact information utilized in that situation. The values of *M*, *P*, and *Q* are also continually maintained as unfactorizable polynomial variables, which are non-linear and hard to reverse, in order to construct *DLCT* using any machine or computer. The random numbers and XOR operators include inventing hidden, unseen, and unchangeable variables for intruders. A Blocker Function produces a unique 32-bit S-Box suitable for the new Magic Serpent Algorithm. A Blocker Function distracts the attacker since it comprises many mathematical random numbers and XOR operators. Additionally, most mathematical XOR operators and random numbers are irreversible. For more mathematical characteristics of a Blocker Function and C++ explanations, refer to Section 4 and Figure 1.

## 4. Research Methodology

The main purpose of this study was to secure Serpent found on IoT devices against DL attacks. The original 4-bit output S-Boxes of Serpent were replaced with new generated 32-bits output S-Boxes. A new mathematical function called Blocker was developed using three magic numbers. A new 32-bit output S-Boxes and Blocker were inserted on Serpent's infrastructure in order to obtain better encryption and decryption processes with resistance to DL attacks. After inserting new 32-bits output S-Boxes and Blocker in Serpent's infrastructure, a newly modified Serpent was developed. In this study, the new modified Serpent, with a new S-Boxes and Blocker, was coined Magic Serpent (Mag_Serpent). The functionality of Mag_Serpent was found to be very different compared to the original Serpent since the encryption process, strength, and the resistance of the DL attacks were stronger than the original Serpent found on IoT devices. The research study was conducted as follows:

1. Serpent was collected from IoT devices (such as smart cards, sensors, and 8-bit processors).
2. The correctness of Serpent was checked and tested by using test vectors given by Serpent developers' reports.
3. All the implemented procedures on Serpent during the process of DL attacks were tested and analyzed using C++.
4. All the original 4-bit output S-Boxes of Serpent were replaced by the newly generated 32-bit output S-Boxes.
5. Three magic numbers were used to generate a new function called Blocker inserted in Serpent infrastructure using C++ implementation. Refer to Figure 1.
6. All functions retrieving S-Boxes of 4-bits output from original Serpent were changed to retrieve a Blocker Function with 32-bit output S-Boxes. Let us examine the following example.

$$Output = S_i(x) \qquad (3)$$

Note: $S_i(x)$ on Equation (3) is 4-bit output S-Box. Equation (3) is replaced to retrieve Equation (4).

$$Blocker(S_i(x), Output) \qquad (4)$$

$S_i(x)$ on Equation (4) is a 32-bits output S-Box since all original 4-bit output S-Boxes were replaced with new 32-bit output S-Boxes. Upon key generation, defined in Figure 3, the Golden ratio $\phi = 9E3779B9$ is also replaced by magic number $M = 4,294,967,296$.

7. The possibility for DL attacks was verified with respect to whether it was still successful after new S-Boxes and Blocker had been applied or inserted. If it was still possible, steps three and four are repeated.
8. If a DL attack was blocked on steps three, four, and five, then a new algorithm inserted with new 32-bit output S-Boxes and Blocker was accepted as a Magic Serpent (Mag_Serpent). As a result, Mag_Serpent was found to be resistant to DL attacks.

The research methodology was conducted on how to make *DLCT* more unmanageable with respect to preventing attackers from discovering Serpent's keys after a DL attack

is applied. It was already stated that the security of Serpent depends on the size of the S-Box output bits. The original output bits of Serpent's S-Boxes were found to be short (4-bit). It is easy for attackers to attack such a type of algorithm. New generated 32-bit output S-Boxes were used to replace all 4-bit output S-Boxes in order to increase the size of output bits so that Serpent is secured against DL attacks. It was found that the new 32-bit output S-Boxes worked successfully in preventing DL attacks, while the Blocker Function worked successfully to block the construction of *DLCT*. The research methodology is summarized using the schematic diagram in Figure 4. The results used successfully to obstruct the construction of *DLCT* and yielded a complicated process for conducting DL attacks on Serpent.

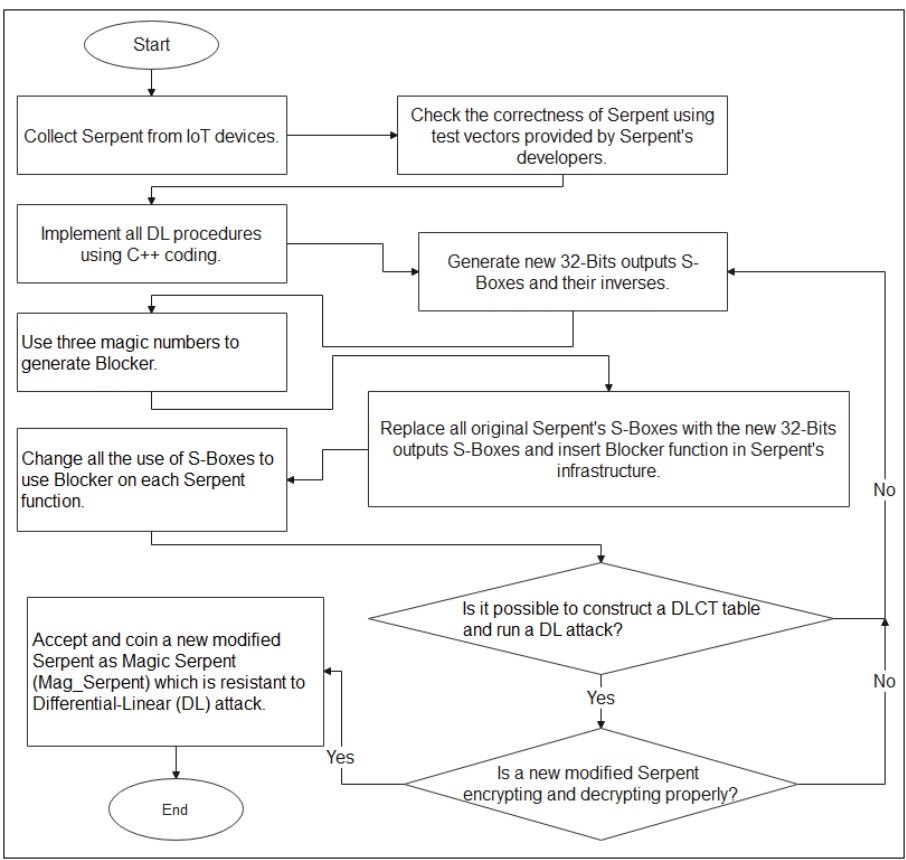

**Figure 4.** Schematic diagram of research methodology.

The Serpent's S-Boxes were found to be $4 \times 4$, meaning that they had 4-bit inputs and 4-bit outputs. It was easy to construct *DLCT* using these kinds of S-Boxes. The *DLCT*s of original Serpent's S-Boxes were tables of $2^4 \times 2^4$ matrix with high-probability elements for discovering secret keys. Generally, if an S-Box has N-bits of inputs and M-bits of output, then its *DLCT* when constructed will be a $2^N \times 2^M$ matrix. Hence, the *DLCT* of the first S-Box of Serpent, defined in Table 1, was said to be $2^4 \times 2^4$. A C++ program code was written to construct *DLCT* of the original first S-Boxes defined in Table 1 using Equation (2). It was proven that it is an easy method to attack Serpent using *DLCT*, as discussed by Bar-On et al. [6] and Canteaut et al. [7]. In order to block the DL attack, the new 32-bit output S-Boxes were generated to replace Serpent's original S-Boxes. For instance, Tables 1–8 were replaced with Tables 10–17, respectively. The Blocker Function was constructed from three magic numbers using C++ code given in Figure 1. The magic numbers were $Q = 4,302,746,963$, $P = 4,559,351,687$, and $M = 4,294,967,296$.

**Table 10.** New generated 32-bit output S-Box to replace Table 1.

| $X$ | $S_0(X)$ | $InvS_0(X)$ |
|---|---|---|
| 0 | 411264f80 | 411264f80 |
| 1 | 91377da1f | 10fc22f87 |
| 2 | 1016b6cf64 | 7128a6f79 |
| 3 | 21038e4da | e15c964be |
| 4 | b146544c5 | a13ee8f72 |
| 5 | 7128a6f79 | f16401a11 |
| 6 | 61213ba26 | 1016b6cf64 |
| 7 | c14dbfa18 | 91377da1f |
| 8 | f16401a11 | 61213ba26 |
| 9 | e15c964be | d1552af6b |
| A | 5119d04d3 | c14dbfa18 |
| B | 310af9a2d | 8130124cc |
| C | 8130124cc | b146544c5 |
| D | 10fc22f87 | 21038e4da |
| E | a13ee8f72 | 5119d04d3 |
| F | d1552af6b | 310af9a2d |

**Table 11.** New generated 32-bit output S-Box to replace Table 2.

| $X$ | $S_1(X)$ | $InvS_1(X)$ |
|---|---|---|
| 0 | 1016b6cf64 | 1016b6cf64 |
| 1 | d1552af6b | b146544c5 |
| 2 | 310af9a2d | 21038e4da |
| 3 | 8130124cc | e15c964be |
| 4 | a13ee8f72 | 61213ba26 |
| 5 | 10fc22f87 | 411264f80 |
| 6 | 61213ba26 | 7128a6f79 |
| 7 | b146544c5 | 10fc22f87 |
| 8 | 21038e4da | 5119d04d3 |
| 9 | c14dbfa18 | a13ee8f72 |
| A | f16401a11 | f16401a11 |
| B | 91377da1f | 8130124cc |
| C | 7128a6f79 | 310af9a2d |
| D | e15c964be | d1552af6b |
| E | 411264f80 | 91377da1f |
| F | 5119d04d3 | c14dbfa18 |

**Table 12.** New generated 32-bit output S-Box to replace Table 3.

| $X$ | $S_2(X)$ | $InvS_2(X)$ |
|---|---|---|
| 0 | 91377da1f | 91377da1f |
| 1 | 7128a6f79 | 1016b6cf64 |
| 2 | 8130124cc | 310af9a2d |
| 3 | a13ee8f72 | a13ee8f72 |
| 4 | 411264f80 | 5119d04d3 |
| 5 | d1552af6b | 21038e4da |
| 6 | b146544c5 | e15c964be |
| 7 | 1016b6cf64 | f16401a11 |
| 8 | e15c964be | c14dbfa18 |
| 9 | 21038e4da | 7128a6f79 |
| A | f16401a11 | 61213ba26 |
| B | 5119d04d3 | 411264f80 |
| C | 10fc22f87 | 8130124cc |
| D | c14dbfa18 | d1552af6b |
| E | 61213ba26 | b146544c5 |
| F | 310af9a2d | 10fc22f87 |

**Table 13.** New generated 32-bit output S-Box to replace Table 4.

| X | $S_3(X)$ | $InvS_3(X)$ |
|---|---|---|
| 0 | 10fc22f87 | 61213ba26 |
| 1 | 1016b6cf64 | 10fc22f87 |
| 2 | c14dbfa18 | 91377da1f |
| 3 | 91377da1f | 411264f80 |
| 4 | d1552af6b | b146544c5 |
| 5 | a13ee8f72 | a13ee8f72 |
| 6 | 7128a6f79 | 8130124cc |
| 7 | 411264f80 | f16401a11 |
| 8 | e15c964be | 310af9a2d |
| 9 | 21038e4da | d1552af6b |
| A | 310af9a2d | c14dbfa18 |
| B | 5119d04d3 | 7128a6f79 |
| C | b146544c5 | 5119d04d3 |
| D | 8130124cc | 1016b6cf64 |
| E | 61213ba26 | e15c964be |
| F | f16401a11 | 21038e4da |

**Table 14.** New generated 32-bit output S-Box to replace Table 5.

| X | $S_4(X)$ | $InvS_4(X)$ |
|---|---|---|
| 0 | 21038e4da | 10fc22f87 |
| 1 | 1016b6cf64 | a13ee8f72 |
| 2 | 91377da1f | b146544c5 |
| 3 | 411264f80 | 8130124cc |
| 4 | d1552af6b | c14dbfa18 |
| 5 | 10fc22f87 | f16401a11 |
| 6 | c14dbfa18 | 7128a6f79 |
| 7 | 7128a6f79 | e15c964be |
| 8 | 310af9a2d | 411264f80 |
| 9 | 61213ba26 | 61213ba26 |
| A | 5119d04d3 | d1552af6b |
| B | b146544c5 | 310af9a2d |
| C | a13ee8f72 | 5119d04d3 |
| D | f16401a11 | 91377da1f |
| E | 8130124cc | 1016b6cf64 |
| F | e15c964be | 21038e4da |

**Table 15.** New generated 32-bit output S-Box to replace Table 6.

| X | $S_5(X)$ | $InvS_5(X)$ |
|---|---|---|
| 0 | 1016b6cf64 | 1552af6b |
| 1 | 61213ba26 | a13ee8f72 |
| 2 | 310af9a2d | 1016b6cf64 |
| 3 | c14dbfa18 | 5119d04d3 |
| 4 | 5119d04d3 | c14dbfa18 |
| 5 | b146544c5 | f16401a11 |
| 6 | a13ee8f72 | 21038e4da |
| 7 | d1552af6b | 310af9a2d |
| 8 | 10fc22f87 | 10fc22f87 |
| 9 | 411264f80 | 411264f80 |
| A | f16401a11 | 7128a6f79 |
| B | 91377da1f | e15c964be |
| C | e15c964be | 61213ba26 |
| D | 7128a6f79 | 91377da1f |
| E | 8130124cc | b146544c5 |
| F | 21038e4da | 8130124cc |

**Table 16.** New generated 32-bit output S-Box to replace Table 7.

| X | $S_6(X)$ | $InvS_6(X)$ |
|---|---|---|
| 0 | 8130124cc | 61213ba26 |
| 1 | 310af9a2d | 91377da1f |
| 2 | d1552af6b | 310af9a2d |
| 3 | 61213ba26 | f16401a11 |
| 4 | 91377da1f | 1016b6cf64 |
| 5 | 5119d04d3 | 7128a6f79 |
| 6 | 7128a6f79 | d1552af6b |
| 7 | c14dbfa18 | 411264f80 |
| 8 | f16401a11 | c14dbfa18 |
| 9 | a13ee8f72 | 5119d04d3 |
| A | 21038e4da | 8130124cc |
| B | 1016b6cf64 | a13ee8f72 |
| C | e15c964be | 21038e4da |
| D | 411264f80 | e15c964be |
| E | b146544c5 | b146544c5 |
| F | 10fc22f87 | 10fc22f87 |

**Table 17.** New Generated 32-bit output S-Box to replace Table 8.

| X | $S_7(X)$ | $InvS_7(X)$ |
|---|---|---|
| 0 | 21038e4da | e15c964be |
| 1 | e15c964be | 411264f80 |
| 2 | 1016b6cf64 | c14dbfa18 |
| 3 | 10fc22f87 | 10fc22f87 |
| 4 | f16401a11 | b146544c5 |
| 5 | 91377da1f | 7128a6f79 |
| 6 | 310af9a2d | 61213ba26 |
| 7 | c14dbfa18 | d1552af6b |
| 8 | 8130124cc | 21038e4da |
| 9 | 5119d04d3 | f16401a11 |
| A | d1552af6b | 5119d04d3 |
| B | b146544c5 | 8130124cc |
| C | a13ee8f72 | 1016b6cf64 |
| D | 411264f80 | a13ee8f72 |
| E | 61213ba26 | 91377da1f |
| F | 7128a6f79 | 310af9a2d |

Tables 10–17 were experimentally written in C++ program to be represented by Figures 5 and 6, where Figure 5 indicated all new 32-bit S-Boxes and Figure 6 indicated all new inverse 32-bit S-Boxes in C++.

*4.1. A Blocker Function*

In this study, a new function described as a Blocker is added (refer to Figure 1). A Blocker Function is an a new generated C++ function implemented solely to develop DL attack blockages on the Serpent algorithm required on IoT devices. This function is developed after the S-Boxes of the Serpent algorithm are transformed to produce 32-bit output S-Boxes. The main purpose of a Blocker Function is to ensure that newly generated 32-bit output S-Boxes suit Serpent's algorithm infrastructure. In simple terms, a Blocker Function regulates all new 32-bit output S-Boxes efficiently utilized throughout the encryption and decryption processes of the newly adjusted Serpent algorithm. A Blocker Function offers a new 32-bit S-Box suitable for the new Magic Serpent Algorithm. A Blocker Function confuses the intruder since it contains many mathematical random numbers. Additionally, most random numbers are irreversible. Without a Blocker Function, a new generated 32-bit output S-Box will not be placed in the algorithms. This Blocker Function has distinct characteristics for ensuring that a DL attack is obstructed. These characteristics are defined as follows:

1.  The output of a Blocker Function is not fixed unlike in S-Boxes where a look-up table is implemented with defined inputs and outputs.
2.  The output of a Blocker Function is secreted and calculated unlike in the Serpent S-Boxes where the output is remarkable on a look-up table.
3.  A Blocker Function is inevitable. If one recognizes an output of a Blocker Function that does not signify an input, it can be reversely estimated and retrieved. The intention is that a Blocker Function is composed of several quantities of random numbers and XOR operators.
4.  Chosen magic numbers (such as *P*, *Q*, and *M*) used in a Blocker Function are unfactor-izable. Refer to Figure 1.
5.  All functions appropriated to comprise a Blocker Function are non-linear.
6.  The input of a Blocker Function is 32-bit long, and the intruder cannot easily create the *DLCT* of $2^{32}$ using a computer or any processor since a lot of memory is required.
7.  A Blocker Function acquires the output of 32-bit S-Boxes and manipulates them as its input. Then, an outstanding output value is produced in order to be utilized in the Magic Serpent algorithm. A new distinct output value is unpredictable; hence, it confuses the intruders.
8.  The output of 32-bit S-Boxes is determined as *state32hold*. A Blocker Function receives this output as its input and returns an unpredictable variable called *statehold*. Refer to Figure 1.
9.  After executing a Blocker Function, all functions in the Serpent algorithm recalling S-Boxes have to identify or employ a Blocker Function because S-Boxes are mathematically preserved and unalterable in a Blocker Function.
10. A Blocker Function provides tamper-proof 32-bit output S-Boxes. Let us suppose that the positions of 32-bit output S-Boxes are altered or the 32-bit S-Boxes are displaced. In that case, Mag_Serpent will not produce the anticipated results.

This study applies a Blocker Function to create a new 32-bit S-Box suitable for the new Magic Serpent Algorithm, and this distracts the attacker since it contains many mathematical random numbers and XOR operators. Additionally, most mathematical random numbers and XOR operators are irreversible. Unlike the traditional S-Boxes employed in Serpent algorithms, a Blocker Function has supplemented robustness against a DL attack. A Blocker Function works favorably in both and is suitable with respect to the new 32-bit S-Boxes and prevents a DL attack of a new Magic Serpent algorithm. Mathematically, a Blocker Function is created as follows.

Assign: $M$ = 4,294,967,296

Perform: Change the value of *state32hold*, using *state32hold* as the value of the following: $state32hold = state32hold \times (\frac{state32hold}{M}) + M \times (\frac{M}{state32hold})$ where *state32hold* is an input of a Blocker Function from 32-bit S-Box.

Check if the value of *state32hold* is greater than *M*. If *state32hold* > *M*, open a first loop of *if* statement. Assign the following:

$Q$ = 4,559,351,687;

$P$ = 4,302,746,963.

*iSecret* is a random number with the range up to a value of ($P \oplus Q$). This number is with the magic numbers in the entire Blocker Function. *iSecret* is a random number, and it is unpredictable and irreversible. Close a first loop of *if* statement.

Check if this the value of *state32hold* is less than or equal to *M*. If *state32hold* < or =*M*, open a second loop of *if* or *else* statement. Assign the following: *iSecret* is a random number with the range up to ($Q \oplus M$). Assign $M = state32hold$ and $Q = state32hold <<< 2$, where $<<<$ is the left round shifting of the number of bits, for example, five in decimal notation = 0101 in binary notation. If 0101 is round left-shifted one (by one), then 0101 will be 1010 in binary notation, which equals 10 in decimal notation or *A* in hexadecimal notation. Therefore, $5 <<< 1 = 10$ is represented in decimal notation. Assign the following:

$P = state32hold <<< 4$;

$Q = M \oplus P$;
$P = M \oplus Q$;
$M = Q \oplus P$.

Change the values of M, P, and Q into random numbers in a range from 0 to iSecret; mathematically, this can be expressed as follows:

$M = rand(M) \ modulo \ iSecret$
$P = rand(P) \ modulo \ iSecret$
$Q = rand(Q) \ modulo \ iSecret$

the where *modulo* operation is the mathematical operator that returns the remainder of a division random number $x$ denoted by $rand(x)$ and iSecret. In this study, $x$ can be $M$, $P$, or $Q$. Close a second loop of *if* or *else* statement.

Recollect all the declared values calculated from the first and second *if* statements. If the recollected values pass a variable $Q$ greater than zero, then create a variable called *TempState*.

Assign the following: $TempState = NOT(state32hold) \ AND \ Q$. where $NOT$ and $AND$ are mathematically bitwise operators. Note that $NOT$ operator returns negative numbers increased by one if an input is a positive integer. For instance, $NOT(5) = -6$, $NOT(10) = -11$, $NOT(2) = -3$, and so on.

Assign the following: $state32hold = |(state32hold \oplus Q|$, where $|x|$ is an absolute operator. An absolute operator converts every negative variable to a positive variable. For instance, $|-y| = |y| = y$.

Assign the following:
$statehold = (\frac{state32hold}{P}) \oplus 187$;
$iSecret = rand(iSecret) \ modulo \ (P \oplus M)$.
Assign the following: $Q = TempState <<< 1$.

Note that the creation of $Q = TempState <<< 1$ always decreases the value of $Q$ continuously until $Q$ is less than zero. A Blocker Function also checks if $Q$ is greater than zero. If $Q$ is more significant than zero, recur the third *for* loop until $Q$ is less than zero or change $Q$, $P$, and $M$ values into random numbers with a range of zero to the value of *statehold*.

$Q = rand(Q) \ modulo \ (statehold)$
$M = rand(M) \ modulo \ (statehold)$
$P = rand(P) \ modulo \ (statehold)$

Transfer or replace the new value of *statehold* that will be used by different Serpent functions or other building blocks used on the Serpent algorithm.

Close the third *for* loop.

Close a Blocker Function.

A Blocker Function uses a 32-bit output value from S-Box as *state32hold* and delivers a new value *statehold* value as an output. A Blocker Function also turns $P$ value, $M$ value, and $Q$ value into unfactorizable polynomials. Random numbers and XOR operators are utilized for complexity and confusion to prevent reverse engineering for attackers. The XOR operator and $rand()$ change the values of the variables inside a Blocker Function. The random numbers and XOR operators also provide a problematic input range when invaders reverse a Blocker Function to calculate the exact information utilized in that situation. The value of $M$, $P$, and $Q$ are also continually maintained as unfactorizable polynomial variables, which are non-linear and hard to reverse, in order to construct *DLCTs* using any machine or computer. The random numbers and XOR operators include inventing hidden, unseen, and unchangeable variables for intruders. A Blocker Function produces a unique 32-bit S-Box suitable for the new Magic Serpent Algorithm. A Blocker Function distracts the attacker since it comprises many mathematical random numbers and XOR operators. Additionally, most mathematical XOR operators and random numbers are irreversible. For more mathematical characteristics of a Blocker Function and C++ explanations, refer to Figure 1. For more detail of a Blocker Function and flowchart, refer to Appendix A Figure A1.

```cpp
//New 4x32 bit Serpent S-boxes//
static const uint64_t S[8][16] = {
{0x411264f80, 0x91377da1f, 0x1016b6cf64,
 0x21038e4da, 0xb146544c5, 0x7128a6f79,
 0x61213ba26, 0xc14dbfa18, 0xf16401a11,
 0xe15c964be, 0x5119d04d3, 0x310af9a2d,
 0x8130124cc, 0x10fc22f87, 0xa13ee8f72,
 0xd1552af6b},
{0x1016b6cf64,
 0xd1552af6b, 0x310af9a2d, 0x8130124cc,
 0xa13ee8f72, 0x10fc22f87, 0x61213ba26,
 0xb146544c5, 0x21038e4da, 0xc14dbfa18,
 0xf16401a11, 0x91377da1f, 0x7128a6f79,
 0xe15c964be, 0x411264f80, 0x5119d04d3},
{0x91377da1f, 0x7128a6f79, 0x8130124cc,
0xa13ee8f72, 0x411264f80, 0xd1552af6b,
0xb146544c5, 0x1016b6cf64, 0xe15c964be,
0x21038e4da, 0xf16401a11, 0x5119d04d3,
0x10fc22f87, 0xc14dbfa18, 0x61213ba26,
0x310af9a2d},
{0x10fc22f87, 0x1016b6cf64, 0xc14dbfa18,
 0x91377da1f, 0xd1552af6b, 0xa13ee8f72,
 0x7128a6f79, 0x411264f80, 0xe15c964be,
 0x21038e4da, 0x310af9a2d, 0x5119d04d3,
 0xb146544c5, 0x8130124cc, 0x61213ba26,
 0xf16401a11},
{0x21038e4da, 0x1016b6cf64, 0x91377da1f,
0x411264f80, 0xd1552af6b, 0x10fc22f87,
0xc14dbfa18, 0x7128a6f79, 0x310af9a2d,
0x61213ba26, 0x5119d04d3, 0xb146544c5,
0xa13ee8f72, 0xf16401a11, 0x8130124cc,
0xe15c964be},
{0x1016b6cf64, 0x61213ba26, 0x310af9a2d,
 0xc14dbfa18, 0x5119d04d3, 0xb146544c5,
 0xa13ee8f72, 0xd1552af6b, 0x10fc22f87,
 0x411264f80, 0xf16401a11, 0x91377da1f,
 0xe15c964be, 0x7128a6f79, 0x8130124cc,
 0x21038e4da},
{0x8130124cc, 0x310af9a2d, 0xd1552af6b,
 0x61213ba26, 0x91377da1f, 0x5119d04d3,
 0x7128a6f79, 0xc14dbfa18, 0xf16401a11,
 0xa13ee8f72, 0x21038e4da, 0x1016b6cf64,
 0xe15c964be, 0x411264f80, 0xb146544c5,
 0x10fc22f87},
{0x21038e4da, 0xe15c964be, 0x1016b6cf64,
 0x10fc22f87, 0xf16401a11, 0x91377da1f,
 0x310af9a2d, 0xc14dbfa18, 0x8130124cc,
 0x5119d04d3, 0xd1552af6b, 0xb146544c5,
 0xa13ee8f72, 0x411264f80, 0x61213ba26,
 0x7128a6f79}};
```

**Figure 5.** New 32-bit S-Boxes written in C++.

```
//New x32 bit Serpent inverse S-boxes//
static const uint64_t IS[8][16] = {
{0x411264f80, 0x10fc22f87, 0x7128a6f79,
 0xe15c964be, 0xa13ee8f72, 0xf16401a11,
 0x1016b6cf64, 0x91377da1f, 0x61213ba26,
 0xd1552af6b, 0xc14dbfa18, 0x8130124cc,
 0xb146544c5, 0x21038e4da, 0x5119d04d3,
 0x310af9a2d},
{0x1016b6cf64, 0xb146544c5, 0x21038e4da,
0xe15c964be, 0x61213ba26, 0x411264f80,
0x7128a6f79, 0x10fc22f87, 0x5119d04d3,
0xa13ee8f72, 0xf16401a11, 0x8130124cc,
0x310af9a2d, 0xd1552af6b, 0x91377da1f,
0xc14dbfa18},
{0x91377da1f, 0x1016b6cf64, 0x310af9a2d,
0xa13ee8f72, 0x5119d04d3, 0x21038e4da,
0xe15c964be, 0xf16401a11, 0xc14dbfa18,
0x7128a6f79, 0x61213ba26, 0x411264f80,
0x8130124cc, 0xd1552af6b, 0xb146544c5,
0x10fc22f87},
{0x61213ba26, 0x10fc22f87, 0x91377da1f,
0x411264f80, 0xb146544c5, 0xa13ee8f72,
0x8130124cc, 0xf16401a11, 0x310af9a2d,
0xd1552af6b, 0xc14dbfa18, 0x7128a6f79,
0x5119d04d3, 0x1016b6cf64, 0xe15c964be,
0x21038e4da},
{0x10fc22f87, 0xa13ee8f72, 0xb146544c5,
0x8130124cc, 0xc14dbfa18, 0xf16401a11,
0x7128a6f79, 0xe15c964be, 0x411264f80,
0x61213ba26, 0xd1552af6b, 0x310af9a2d,
0x5119d04d3, 0x91377da1f, 0x1016b6cf64,
0x21038e4da},
{0xd1552af6b, 0xa13ee8f72, 0x1016b6cf64,
0x5119d04d3, 0xc14dbfa18, 0xf16401a11,
0x21038e4da, 0x310af9a2d, 0x10fc22f87,
0x411264f80, 0x7128a6f79, 0xe15c964be,
0x61213ba26, 0x91377da1f, 0xb146544c5,
0x8130124cc},
{0x61213ba26, 0x91377da1f, 0x310af9a2d,
0xf16401a11, 0x1016b6cf64, 0x7128a6f79,
0xd1552af6b, 0x411264f80, 0xc14dbfa18,
0x5119d04d3, 0x8130124cc, 0xa13ee8f72,
0x21038e4da, 0xe15c964be, 0xb146544c5,
0x10fc22f87},
{0xe15c964be, 0x411264f80, 0xc14dbfa18,
0x10fc22f87, 0xb146544c5, 0x7128a6f79,
0x61213ba26, 0xd1552af6b, 0x21038e4da,
0xf16401a11, 0x5119d04d3, 0x8130124cc,
0x1016b6cf64, 0xa13ee8f72, 0x91377da1f,
0x310af9a2d}};
```

**Figure 6.** New inverse of 32-bit S-Boxes written in C++.

*4.2. Experimental Confirmation of DL Attack on Serpent*

This study experimentally verified and analyzed the DL attack conducted in [19] on a 12-round Serpent. The attack was based on the fundamental 11-round DL attack using a plaintext pair that provides the input differentials of 28 participating S-Boxes in round zero. Consequently, changing the Serpent algorithm was conceivable, and a 12-round attack against Serpent with 256-bit keys was obtained.

Dunkelman et al. [19] tried all the possible input differences for round 1 that yielded the difference $LT^{-1}(\Delta P) = 20000000000001A00E00400000000000_x$. The difference was not changed by S-Boxes 2, 3, 19, 23, and so on; those S-Boxes did not change the participating bits of $LT^{-1}(\Delta P)$ [19]. Consequently, Dunkelman et al. [19] constructed plaintext structures, which took that fact into attention and obtained a 12-round attack on Serpent:

1.  Dunkelman et al. [19] selected $N = 2^{123.5}$ plaintexts that consisted of $2^{11.5}$ structures, and each was selected by choosing the following: (a) an abitaray plaintext $P_0$; (b) the plaintexts $P_1, \ldots, P_{2^{112}-1}$, which differed from $P_0$ by all the $2^{112} - 1$ possibilities of non-empty subsets of the bits which were used as inputs of all S-Boxes except 2, 3, 19, and 23 in round zero [19].
2.  Dunkelman et al. [19] requested the cipher texts of the encrypted plaintext structures by using the private unknown key $K$. 3. For every input 112-bit of $K_0$ value using those 28 S-Boxes, partly encrypted all the plaintexts in the first round and utilize the original 11-round DL attack on Serpent [19].
3.  Each experimental key revealed and provided Dunkelman et al. [19] 112 + 20 + 28 = 160-bit subkeys: 112-bit of round 0; 20-bit of round 1; and 28-bit of round 11, simultaneously with an accuracy test [19]. The accurate estimation of the 160-bit was anticipated to be the typical and frequently expected value with the appearance of more than 84% completion rate [19].
4.  The remainder of the key bits were retrieved by supplemental techniques [19].

The study experimentally verified that the data attack complexity was $2^{123.5}$ chosen plaintexts. The time attack complexity $2^{123.5} \times (\frac{28}{384}) \times 2^{112} = 2^{231.7}$ encryptions for the partial encryption in Step 3, and $2^{137.4} \times 2^{112} = 2^{249.4}$ for the repeated trials of the 11-round DL attack [19].

The study further experimentally verified that on a 10-round DL attack of Serpent using 128-bit keys, the data complexity was $2^{101.2}$ elected plaintexts, and time encryption complexity was $2^{115.2}$.

*4.3. Experimental Contribution of DL Attack on a Newly Generated Mag_Serpent*

Mag_Serpent used a new 32-bit S-Box, which declined to execute C++ *DLCT* from various computers and machines due to memory limitations on diverse computers and machines. No computers and machines could compute the *DLCT* of $2^4 \times 2^{32}$ = 16 × 429,4967,296 matrix, which is presumed to carry 68,719,476,736 entities. Without *DLCT*, it was impracticable to conduct a DL attack on a newly generated 4 × 32 S-Boxes of Mag_Serpent algorithm. No rounds out of 32 were attacked using the DL attack due to the new 32-bit output S-Boxes, which obstructed the development of the *DLCT* due to memory constraints.

A review of how *DLCT* was theoretically developed was examined and programmed experimentally in C++ code for validation, testing, confirmation, and verification. On the Serpent, the results revealed that the DL attack was possible. The main building blocks that performed all possibilities of the DL attack were the size of the S-Boxes. The Serpent's S-Boxes were 4 × 4, indicating 4-bit inputs and 4-bit outputs. The experiment determined that it was straightforward to build *DLCT* utilizing the 4 × 4 Serpent S-Boxes (refer to Table 9 and Figure 7).

**Figure 7.** C++ experimental results of *DLCT*

The experiment used a C++ program to generate the *DLCT* of 4 × 4 and 4 × 32 S-Box. The code validation was examined by using a 4 × 4 Serpent S-Boxes and a newly generated 4 × 32 S-Box of Mag_Serpent algorithm. The purpose of validating the code was to confirm the correctness of the written C++ experimental output *DLCT* compared to the theoretical outputs. Note that the *DLCT* of 4 × 4 S-Box is a matrix of $2^4 × 2^4 = 16 × 16$ matrix with 256 entities (refer to Table 9 and Figure 7).

The experiment continued on a newly developed 4 × 32 S-Box of Mag_Serpent algorithm. The program malfunctioned after five hours before the *DLCT* was executed. No computer or machine could compute the *DLCT* of $2^4 × 2^{32} = 16 × 4,294,967,296$ matrix, which is expected to contain 68,719,476,736 entities. Without the *DLCT*, conducting a DL attack on a newly developed 4 × 32 S-Box of Mag_Serpent algorithm was impracticable (refer to Table 9 and Figure 7).

*DLCT* of 4 × 4 S-Box had the first integer 16, which is $(2^4)$ considering the S-Box required four bits output as the most distinguished parameter. Sixteen is a byte that was donated as 00010000 in binary notation. If each 4 × 4 S-Box *DLCT* was treated as a byte, then the memory required to construct 4 × 4 S-Box *DLCT* is 8 bits × 256 = 256 bytes. Note that 256 is the number of items displayed on a 4 × 4 S-Box *DLCT*. A machine or computer can efficiently handle 4096 bytes (refer to Table 9 and Figure 7).

From the above computations, the S-Box required thirty-two bits as the first parameter. The study presumed that the *DLCT* of 4 × 32 S-Box would have the first number item as 4,294,967,296, which is $(2^{32})$. The 4,294,967,296 number is a triple-word comprising 5 bytes donated as 00000000100000000000000000000000000000000 in binary notation. If each 4 × 32 S-Box *DLCT* element were treated as a triple-word, then the memory required to build 4 × 32 S-Box *DLCT* would be 40 *bits* × $2^4 × 2^{32} = 343,597,383,680$ bytes. Note that 343,597,383,680 was an expected number of entities displayed on a 4 × 32 S-Box *DLCT*. A machine or computer could not easily handle a computation memory of 343,597,383,680 bytes of each item. Hence, C++ *DLCT* of the 4 × 32 S-Box program malfunctioned before execution (refer to Table 9 and Figure 7).

*DLCT* of the Serpent S-Box was a table with $2^4$ rows × $2^4$ columns with high probabilities of comprehending a key. The experiment used the C++ program to generate the *DLCT* of 4 × 4 Serpent S-Box. After examining the method, the results confirmed that attacking the Serpent algorithm using *DLCT* was achievable. The study applied the newly created 32-output-bit S-Boxes on Serpent found on IoT devices in order to block a DL attack (refer to Table 9 and Figure 7). Table 9 was presumed *DLCT*, and Figure 7 was the experimentally analyzed *DLCT* performed by running a C++ *DLCT* code. A C++ *DLCT* code was used to

prove and confirm that the study of building a *DLCT* was conducted with all methods of a DL attack on a Serpent.

The code was also implemented in both Serpent and Mag_Serpent in order to examine whether a DL attack was possible. All results were presented, and the results completely explain the development of the *DLCT* that came before; after a novel approach of utilizing 32 bits, S-Boxes were implemented. The study employed a Blocker Function to create a new 32-bit S-Box suitable for the new Mag_Serpent algorithm and distracted the intruder. The random numbers and XOR operators also provide a problematic input range when attackers reverse a Blocker Function to determine accurate information used in that situation. The values of *M*, *P*, and *Q* are also continually maintained as unfactorizable polynomial variables, which are non-linear and hard to reverse, in order to construct *DLCT* using any machine or computer. The random numbers and XOR operators include inventing hidden, unseen, and unchangeable variables for intruders. A Blocker Function produces a unique 32-bit S-Box suitable for the new Magic Serpent Algorithm. A Blocker Function distracts the attacker since it comprises many mathematical random numbers and XOR operators. Additionally, most mathematical XOR operators and random numbers are irreversible. For more mathematical characteristics of a Blocker Function and C++ explanations, refer to Section 4 and Figure 1.

In this study, Mag_Serpent was resistant to a DL attack and created a new $4 \times 32$ S-Box. The study used a Blocker Function to insert the new 32-bit S-Boxes that are suitable for the new Mag_Serpent algorithm. The study used a Blocker Function to confuse the attacker since it comprises many mathematical random numbers and XOR operators. Additionally, most mathematical XOR operators and random numbers are irreversible. The new Mag_Serpent successfully decrypted and encrypted information after adopting a Blocker Function and the new $4 \times 32$ S-Boxes. The code of the new Mag_Serpent algorithm is obtainable upon request. The C++ code confirmed that a DL attack was permissible to a standard Serpent on several rounds, including round 12 before applying a Blocker Function and the new $4 \times 32$ S-Boxes. Nevertheless, after applying a Blocker Function and the new $4 \times 32$ S-Boxes as a novelty, the C++ code confirmed that the study blocked the DL attack successfully on Mag_Serpent. Additionally, creating a *DLCT* matrix with $2^{32}$ rows and columns was not straight forward due to the memory constraints of the computer.

## 5. Results and Analysis

On Serpent, the results revealed that the DL attack was possible. The main building blocks that performed all possibilities of the DL attack were the size of the S-Boxes. The Serpent's S-Boxes were $4 \times 4$, indicating 4-bit inputs and 4-bit outputs. The experiment determined that it was straightforward to build a *DLCT* utilizing $4 \times 4$ Serpent S-Boxes (refer to Table 9 and Figure 7).

The experiment used a C++ program to generate a *DLCT* of $4 \times 4$ and $4 \times 32$ S-Box. The validation of the code was examined by using a $4 \times 4$ Serpent S-Box and a new generated $4 \times 32$ S-Box of Mag_Serpent algorithm. The purpose of validating the code was to confirm the correctness of the written C++ experimental output *DLCT* compared to the theoretical outputs. Note that the *DLCT* of $4 \times 4$ S-Box is a matrix of $2^4 \times 2^4 = 16 \times 16$ matrix with 256 entities (refer to Table 9 and Figure 7).

The experiment continued on a newly developed $4 \times 32$ S-Box of Mag_Serpent algorithm. The program malfunctioned after five hours before *DLCT* was executed. No computer or machine could compute the *DLCT* matrix of $2^4 \times 2^{32} = 16 \times 4,294,967,296$, which is expected to contain 68,719,476,736 entities. Without the *DLCT*, it was impracticable to conduct a DL attack on a newly developed $4 \times 32$ S-Box of Mag_Serpent algorithm. On Serpent, results revealed that the DL attack was possible. The main building blocks that performed all possibilities of the DL attack were the size of the S-Boxes. The Serpent's S-Boxes were $4 \times 4$, indicating 4-bit inputs and 4-bit outputs. The experiment determined that it was straightforward to build a *DLCT* utilizing the $4 \times 4$ Serpent S-Boxes (refer to Table 9 and Figure 7).

A *DLCT* of 4 × 4 S-Box has a first integer of 16, which is ($2^4$) considering the fact that S-Box required four-bit outputs as the most distinguished parameter. Sixteen is a byte donated as 00010000 in binary notation. If each 4 × 4 S-Box *DLCT* is treated as a byte, then the memory required to construct a 4 × 4 S-Box *DLCT* was 8-bit × 256 = 256 bytes. Note that 256 is the number of items displayed on a 4 × 4 S-Box *DLCT*. A machine or computer can efficiently handle 4096 bytes (refer to Table 9 and Figure 7).

From the above computations, S-Box required thirty-two bits as the first parameter. The study presumed that the *DLCT* of 4 × 32 S-Box would have the first number item as 4,294,967,296, which is ($2^{32}$). The number 4,294,967,296 is a triple word comprising 5 bytes donated as 0000000010000000000000000000000000000000 in binary notation. If each 4 × 32 S-Box *DLCT* element was treated as a triple-word, then the memory required to build 4 × 32 S-Box *DLCT* would be 40 *bits* × $2^4$ × $2^{32}$ = 343,597,383,680 bytes. Note that 343597383680 was an expected number of entities displayed on a 4 × 32 S-Box *DLCT*. A machine or computer cannot easily handle a computation memory of 343597383680 bytes of each item. Hence, the C++ *DLCT* of the 4 × 32 S-Box program malfunctioned before execution. Refer to Table 9 and Figure 7.

*DLCT* of the Serpent S-Box was a table of $2^4$ rows × $2^4$ columns with high probabilities of comprehending a key. The experiment used the C++ program to generate the *DLCT* of 4 × 4 Serpent S-Box. After examining the method, the results confirmed that attacking the Serpent algorithm using *DLCT* was achievable. The study applied the newly created 32-output-bit S-Boxes on Serpent found on IoT devices to block a DL attack (refer to Table 9 and Figure 7).

The new 32-bit output S-Boxes prevented the construction of *DLCT*, which was presumed to be a $2^4$ × $2^{32}$ matrix, since the new output bits increased from 4 bit to 32 bit. That is, a $2^4$ × $2^{32}$ = 256 × 4,294,967,296 matrix is required to construct *DLCT*, which needs high computer memory in order to compute and display such a matrix. The experiment demonstrated that it was impracticable to construct a *DLCT* of a new 32-bit output S-Box using Equation (2) if the Blocker Function is embedded on Serpent's structure, since the maximum size limitation was limited and the memory required had been exceeded. The C++ program of the *DLCT* with respect to new S-Boxes clashed before *DLCT* was finally constructed due to the memory limitation of a computer. An ordinary computer cannot construct a matrix of 256 columns × 4,294,967,296 rows. The experiment also confirmed that it was impractical to construct a matrix of 256 × 4,294,967,296, since a computer has a maximum memory of $2^{64}$, which is impossible. If ther eis no *DLCT*, there will be no DL ttacks. Therefore, in this study, securing Serpent against DL attacks by using the Blocker Function and 32-bit output S-Boxes worked successfully. The experiment showed that when the Blocker Function was inserted in Serpent's infrastructure, all positions of 32-bit S-Boxes were unchangeable. For instance, a new 32-bit output $S_0(X)$ cannot be changed nor substituted with any arbitrarily new 32-bit output S-Box such as $S_1(X)$, $S_2(X)$, ..., or $S_7(X)$. The newly generated S-Boxes cannot be replaced by any 32-bit output S-Box taken from other known algorithms, even though the sizes are equal.

In this study, all procedures used to attack the original Serpent using DL attacks were studied and conducted. The C++ programs were written to confirm if an original Serpent could be attacked using *DLCT* and DL attacks. The C++ programs confirmed and executed the same results defined and provided in Table 9. Table 9 was the theoretical results found by Bar-On et al. [6] when the original Serpent was attacked using 4-bit output S-Boxes defined in Table 1. All procedures used to attack an original Serpent by Bar-On et al. [6] were conducted using C++ programs. In this study, the C++ programs confirmed and validated the theoretical results defined in [6]. The experimental results found in this study are given in Figure 7. Table 9 and Figure 7 had the same elements. Table 9 provides the theoretical *DLCT*, as explained by Bar-On et al. [6], and Figure 7 provides the experimental *DLCT* results conducted in this study. The serpent was attacked on rounds 10 and 11, whereas Mag_Serpent resisted being attacked all rounds during the DL attack process (refer to Table 18 and Figure 8).

**Table 18.** Results of DL Attack.

| Name of Algorithm | Time Complexity | Data Complexity | Rounds Attacked |
|---|---|---|---|
| Serpent | $2^{115.5}$ | $2^{101.2}$ | 10 |
| Serpent | $2^{231.7}$ | $2^{249.4}$ | 11 |
| Mag_Serpent | ∞ | ∞ | 0 |

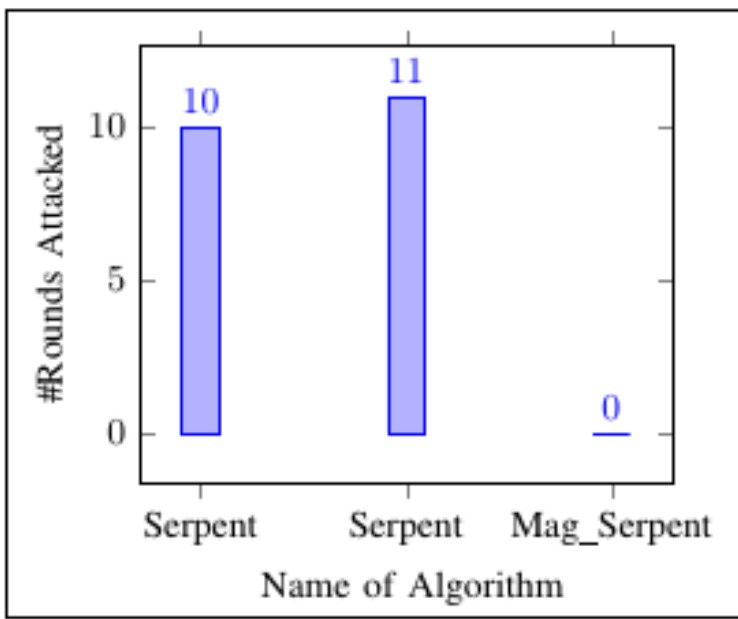

**Figure 8.** Results of DL attack.

The C++ experiment showed that a DL attack was possible with respect to the original Serpent before new S-Boxes and Blocker approaches were implemented, but after the implementation of the novelty of using new 32-bit output S-Boxes and Blocker Function, the DL attack was blocked on a new modified Serpent called Mag_Serpent (refer to Tables 19 and 20).

**Table 19.** Results of feasibility of constructing *DLCT* before and after 32-bit output S-Boxes and Blocker were applied.

| Name of Algorithms | Before 32-Bit Output S-Boxes and Blocker Were applied | After 32-Bit Output S-Boxes and Blocker Were Applied |
|---|---|---|
| Serpent | Construction of *DLCT* was feasible | Construction of *DLCT* was infeasible due to the requirement of memory |

**Table 20.** Results of key discovery before and after 32-bit output S-Boxes and Blocker were Applied.

| Name of Algorithms | Before 32-Bit Output S-Boxes and Blocker Were Applied | After 32-Bit Output S-Boxes and Blocker Were Applied |
|---|---|---|
| Serpent | The key was revealed in all rounds | No discovery of a key was found since no *DLCT*, no DL attack |

In cryptography, the Avalanche Effect is a satisfactory characteristic of algorithms [21]. If one input bit is inverted (flipped), the output bits have to improve significantly. Such a small adjustment in either the plaintext or the key should produce an excessive variation in the ciphertext in strong algorithms [21]. The Avalanche Effect is advanced in order to

obtain a method called the Strict Avalanche Criterion (SAC) for examining the encryption robustness of an algorithm [22]. SAC is achieved if a particular input bit, either the plaintext or the key, returns the transformation of ciphertext output bits of 50% probability [22]. The experiment utilized the Avalanche Effect on Serpent and Mag_Serpent in order to obtain SAC. The results showed that the Serpent and a newly generated Mag_Serpent algorithm had better SAC characteristics. The Avalanche Effect of Mag_Serpent and Serpent on both key and plaintext was approximately 50% probability compared to SAC characteristics (refer to Table 21 and Figures 9–14).

**Figure 9.** Experimental Avalanche Effect of Serpent when one bit of a key was flipped.

```
One changed
63.000000
ffffffffffffffffffffffffffffbfff
One changed
63.000000
ffffffffffffffffffffffffffffdfff
One changed
56.000000
ffffffffffffffffffffffffffffefff
One changed
64.000000
ffffffffffffffffffffffffffff7ff
One changed
72.000000
ffffffffffffffffffffffffffffbff
One changed
53.000000
ffffffffffffffffffffffffffffdff
One changed
65.000000
ffffffffffffffffffffffffffffeff
One changed
64.000000
ffffffffffffffffffffffffffff7f
One changed
59.000000
ffffffffffffffffffffffffffffbf
One changed
57.000000
ffffffffffffffffffffffffffffdf
One changed
67.000000
ffffffffffffffffffffffffffffef
One changed
64.000000
fffffffffffffffffffffffffffff7
One changed
76.000000
fffffffffffffffffffffffffffffb
One changed
64.000000
fffffffffffffffffffffffffffffd
One changed
61.000000
fffffffffffffffffffffffffffffe
One changed
68.000000

Avarage of bits changed out of 128 = 64.492188

Avalanche Effect in Percentage=
= ((Avarage of bits changed out of 128 )/128) x 100 )
= 50.384521

--------------------------------
Process exited after 0.4659 seconds with return value 0
Press any key to continue . . .
```

**Figure 10.** Experimental Avalanche Effect of Serpent when one bit of a plaintext was flipped.

```
One changed
60.000000
0123456789abcdeffedcba98765632100123456789abcdeffedcba9876553210
One changed
77.000000
0123456789abcdeffedcba98765532100123456789abcdeffedcba987654b210
One changed
66.000000
0123456789abcdeffedcba987654b2100123456789abcdeffedcba9876547210
One changed
60.000000
0123456789abcdeffedcba98765472100123456789abcdeffedcba9876541210
One changed
60.000000
0123456789abcdeffedcba98765412100123456789abcdeffedcba9876542210
One changed
59.000000
0123456789abcdeffedcba98765422100123456789abcdeffedcba9876543a10
One changed
67.000000
0123456789abcdeffedcba9876543a100123456789abcdeffedcba9876543610
One changed
73.000000
0123456789abcdeffedcba98765436100123456789abcdeffedcba9876543010
One changed
77.000000
0123456789abcdeffedcba98765430100123456789abcdeffedcba9876543310
One changed
67.000000
0123456789abcdeffedcba98765433100123456789abcdeffedcba9876543290
One changed
63.000000
0123456789abcdeffedcba98765432900123456789abcdeffedcba9876543250
One changed
67.000000
0123456789abcdeffedcba98765432500123456789abcdeffedcba9876543230
One changed
62.000000
0123456789abcdeffedcba98765432300123456789abcdeffedcba9876543200
One changed
61.000000
0123456789abcdeffedcba98765432000123456789abcdeffedcba9876543218
One changed
71.000000
0123456789abcdeffedcba98765432180123456789abcdeffedcba9876543214
One changed
64.000000
0123456789abcdeffedcba98765432140123456789abcdeffedcba9876543212
One changed
57.000000
0123456789abcdeffedcba98765432120123456789abcdeffedcba9876543211
One changed
67.000000

Avarage of bits changed out of 256 = 64.683594

Avalanche Effect in Percentage=
= ((Avarage of bits changed out of 256 )/128) x 100 )
 = 50.534058

--------------------------------
Process exited after 0.9102 seconds with return value 0
Press any key to continue . . .
```

**Figure 11.** Experimental Avalanche Effect of Mag_Serpent when one bit of a key was flipped.

**Figure 12.** Experimental Avalanche Effect of Mag_Serpent when one bit of a plaintext was flipped.

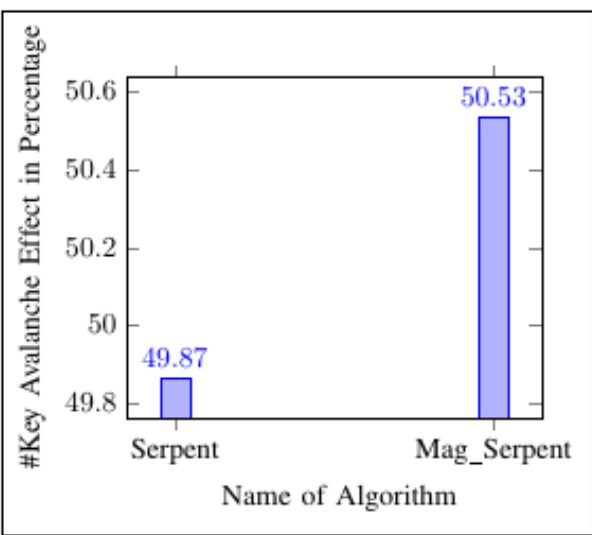

**Figure 13.** Experimental key Avalanche Effect in percentage.

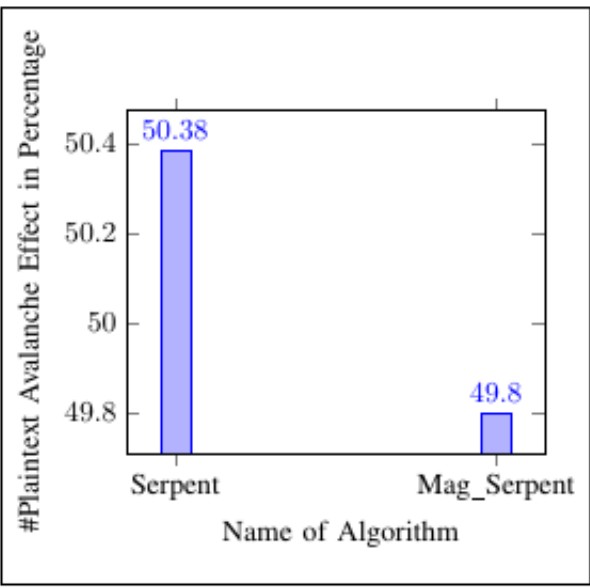

**Figure 14.** Experimental plaintext Avalanche Effect in percentage.

**Table 21.** Avalanche Effect of key and plaintext when one bit was flipped.

| Name of Algorithm | Key Avalanche Effect in Percentage | Plaintext Avalanche Effect in Percentage |
|---|---|---|
| Serpent | 49.8657 | 50.3842 |
| Mag_Serpent | 50.5340 | 49.7985 |

In cryptography, the memory required to install an algorithm is one of the most needed parameters before installation. If an algorithm requires higher memory than the platform or environments installed, that algorithm is neglected irrespective of encryption strength. The study measured the memory of both Serpent and Mag_Serpent using the C++ program. The results showed a memory of 11181 bytes and 13206 bytes for Serpent and Mag_Serpent, respectively (refer to Table 22 and Figures 15–17).

**Table 22.** Memory required for installation of algorithms.

| Name of Algorithm | Memory Required in Bytes |
|---|---|
| Serpent | 11,181 |
| Mag_Serpent | 13,206 |

```
Enter the file name: Serpent.cpp
Size of file: 11181 Bytes
--------------------------------
Process exited after 12.33 seconds
Press any key to continue . . .
```

**Figure 15.** Experimental memory of Serpent for installation.

```
Enter the file name: 32SERPENT.CPP
Size of file: 13206 Bytes
--------------------------------
Process exited after 11.53 seconds with
Press any key to continue . . .
```

**Figure 16.** Experimental memory of Mag_Serpent for installation.

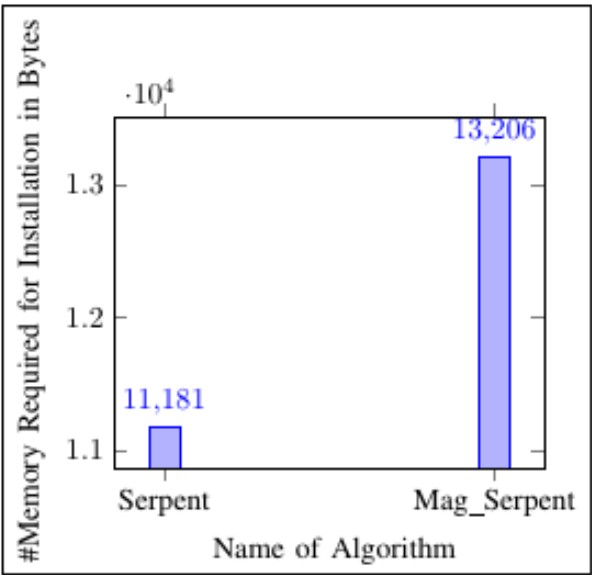

**Figure 17.** Memory required for installation in bytes.

Encryption and decryption of Serpent and Mag_Serpent were experimentally conducted to verify if both algorithms functioned splendidly for the encryption and decryption. The study used an image to test the encryption and decryption of Serpent and Mag_Serpent using the C++ program. The results demonstrated that both encryption and decryption of Serpent and Mag_Serpent were working as expected (refer to Figure 18).

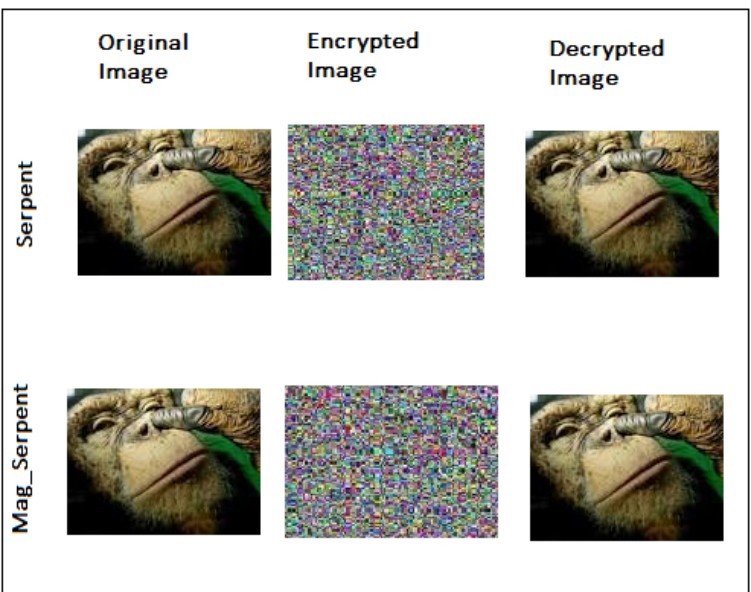

**Figure 18.** Encryption and decryption of image by using Serpent and Mag_Serpent.

## 6. Conclusions and Future Work

The study measured the memory of both Serpent and Mag_Serpent using the C++ program. The results demontrate a memory of 11,181 bytes and 13,206 bytes for Serpent and Mag_Serpent, respectively.

The experiment conducted the Avalanche Effect on Serpent and Mag_Serpent in order to obtain SAC. The results demonstrated that the Serpent and a newly generated Mag_Serpent algorithm had better SAC characteristics. The Avalanche Effect of Mag_Serpent and Serpent on both key and plaintext was approximately 50% probability compared to SAC characteristics.

The experiment continued on a newly developed 4 × 32 S-Box of Mag_Serpent algorithm. The program malfunctioned after five hours before *DLCT* was executed. No computer or machine could compute the *DLCT* of $2^4 \times 2^{32}$ = 16 × 4,294,967,296 matrix, which is expected to contain 68,719,476,736 entities. Without *DLCT*, it was impracticable to conduct a DL attack on a newly developed 4 × 32 S-Box of Mag_Serpent algorithm. On the Serpent, the results revealed that the DL attack was possible. The main building blocks that performed all possibilities of the DL attack were the size of the S-Boxes. The Serpent's S-Boxes were 4 × 4, indicating 4-bit inputs and 4-bit outputs. The experiment determined that it was straightforward to build a *DLCT* utilizing 4 × 4 Serpent S-Boxes.

The C++ experiment showed that a DL attack was possible relative to an original Serpent before new S-Boxes and Blocker approaches were implemented, but after the implementation of the novelty of using new 32-bit output S-Boxes and Blocker Function, the DL attack was blocked on a new modified Serpent called Mag_Serpent.

The study showed that the Serpent algorithm used to secure data stored on IoT devices was secured against DL attacks by using magic numbers and the Blocker Function. It has been confirmed that it is impossible to draw a *DLCT* of 32-bit output S-Box. Furthermore, it has been proven that if the construction of *DLCT* is infeasible on a particular algorithm, then no DL attack will be possible on that algorithm. In this study, a new modified Serpent was generated and named Mag_Serpent.

Future studies will include measuring the power consumption of Mag_Serpent compared to an original Serpent. Other attacks (such as Boomerang, man-in-the-middle, and Denial of Services (DoS)) will be analyzed using a Blocker Function and 32-bit S-Boxes.

**Funding:** This research received no external funding.

**Data Availability Statement:** Not Applicable, the study does not report any data.

**Conflicts of Interest:** The authors declare no conflict of interest.

## Appendix A

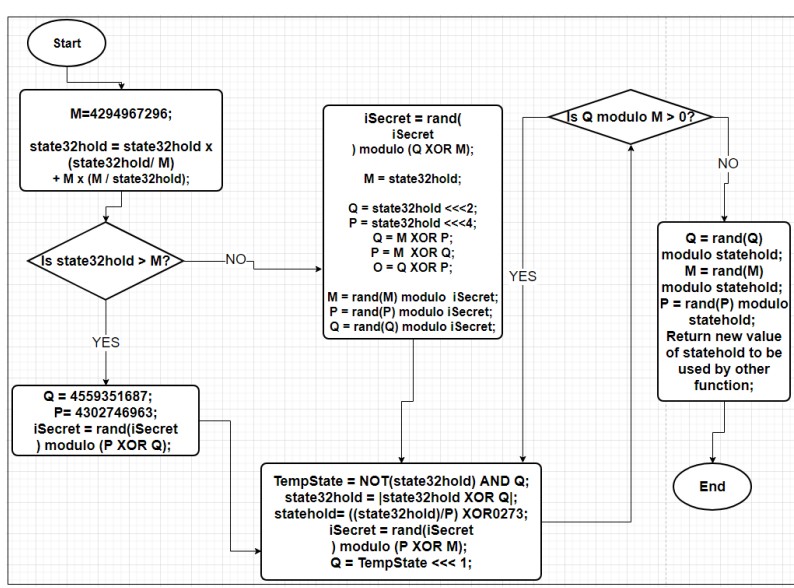

**Figure A1.** Flowchart of a Blocker Function.

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
