# Peer review of "Securing IoT Devices against Differential-Linear (DL) Attack Used on Serpent Algorithm"

_futureinternet, doi:10.3390/fi14020055_

Round 1
Reviewer 1 Report
This paper deals with a securing IoT devices against differential-linear (DL) attack used on serpent algorithm. I would like to point put or comment following as:
- In introduction, description of object in this paper is very ambiguous. Thus, authors should revise it to represent clear object of this paper.
- The description of previous work also adds their advantage and disadvantage which related to the object of this paper
- What is the differentiation compared to previous works? What is novelty? It is not clear.
- Figure 1 and Figure 2 should change the algorithm's description type.
- Please add when someone attacks your algorithm, how to prove protect your algorithm.
Reviewer 2 Report
This manuscript addresses a cybersecurity question related to the adoption of the Serpent cypher in the Internet of Things. With the aim of strengthening Serpent, the Authors propose to integrate it with magic numbers and a function called Blocker. Results of tests are presented and discussed to highlight the performance of the modified Serpent.
In general, I think that this manuscript addresses a relevant topic. However, the quite weak introduction and limited review of related literature do not allow to appreciate the extend and originality of the results that are presented.
Major modifications should thus be made in order to try to bring the work to an acceptable level of quality.
To this end, please consider the comments below.
- In my opinion, the first section is not so effective when trying to introduce the topic addressed in the paper. The Authors should try to highlight more clearly the security issues associated with IoT, trying to illustrate them at a more high level that could be accessible also to readers not so familiar with the topic.
- Focusing again on the introduction, the Authors should better highlight the original contributions of the work. To this end, I would suggest to the Authors to move the section devoted to the review of related literature after the first section: this would allow to better discuss the limits of current approaches and to better identify and discuss the original contributions of this work.
- In order to improve the accessibility of the introductive section for a reader who is not familiar with the topic but interested in the proposed algorithmic contributions, it would be useful to introduce Serpent in one single statement.
- The algorithmic rigor of the Sections presenting the features of the modified Serpent should be enhanced by relying more on the use of pseudocode environment, accompanied by proper exhaustive discussion in the text.
- The quality of all the figures (e.g., Figures 1 and 2) is low and should be checked and improved.
- Section 6 is too short and should be enlarged, better discussing the merits and the drawbacks of the new improved Serpent algorithm. Directions for future research should also be better addressed.
Author Response
"Please see the attachment."

Reviewer 3 Report
Dear authors, while the presentation is nice in shape, there are few comments and/or suggestions to improve the manuscript:
- The significance of the study is not clear to me and there is a serious literature review gap in this paper. I strongly consider that the distinguished authors did not pay attention to the other relevant references. For a review paper, more than 50 references must be used.
- Clarify better the advantages of this paper in the introduction section because in the literature a lot of papers consider the same proposed approach. I strongly recommend the authors have a separate section for the literature review and to discuss the drawbacks of existing works.
- Multiple citation references are used in the wrong way (see refs. 1-21, 21-23..etc)..
- The authors must present the limitation of the considered method.
- The Results and Analysis section should be revised to improve the impact of the paper. The results are abundant but the results’ analysis is not enough. However, I cannot see deeply analysis related to them and cannot understand the meaning of results. Please add more analysis.
- A lot of Figures are unintelligible. Please revise them.
However, the theoretic background is solid and the article is enhanced with very interesting results depicted in the last section. The conclusions section sums up the research output while the readers can find rich information for further study.
Author Response
"Please see the attachment."

Round 2
Reviewer 1 Report
This paper is well revised according to the reviewer's point out. Thus, I would like to decide as an "accept"
Reviewer 2 Report
The Authors have addressed my comments and some passages have been improved.
The manuscript could be now considered for possible acceptance in this journal.
However, the quality of some figures (e.g., 1 and 2) and tables (e.g., 9) is still low and should be improved.
Reviewer 3 Report
The reviewer has no further comments.